# CLASS-INCREMENTAL LEARNING WITH REPETITION

## ABSTRACT

Real-world data streams naturally include the repetition of previous concepts. From a Continual Learning (CL) perspective, repetition is a property of the environment and, unlike replay, cannot be controlled by the user. Nowadays, Class-Incremental scenarios represent the leading test-bed for assessing and comparing CL strategies. This family of scenarios is very easy to use, but it never allows revisiting previously seen classes, thus completely disregarding the role of repetition. We focus on the family of Class-Incremental with Repetition (CIR) scenarios, where repetition is embedded in the definition of the stream. We propose two stochastic scenario generators that produce a wide range of CIR scenarios starting from a single dataset and a few control parameters. We conduct the first comprehensive evaluation of repetition in CL by studying the behavior of existing CL strategies under different CIR scenarios. We then present a novel replay strategy that exploits repetition and counteracts the natural imbalance present in the stream. On both CIFAR100 and TinyImageNet, our strategy outperforms other replay approaches, which are not designed for environments with repetition.

## 1 INTRODUCTION

Continual Learning (CL) requires a model to learn new information from a stream of experiences presented over time, without forgetting previous knowledge (Parisi et al., 2019; Lesort et al., 2020). The nature and characteristics of the data stream can vary a lot depending on the real-world environment and target application. Class-Incremental (CI) scenarios (Rebuffi et al., 2017) are the most popular ones in CL. CI requires the model to solve a classification problem where new classes appear over time. Importantly, when a set of new classes appears, the previous ones are never seen again. However, the model still needs to correctly predict them at test time. Conversely, in a Domain-Incremental (DI) scenario (van de Ven & Tolias, 2019) the model sees all the classes at the beginning and continue to observe new instances of the classes over time.

The CI and DI scenarios have been very helpful to promote and drive CL research in the last few years. However, they strongly constrain the properties of the data stream in a way that it sometimes considered unrealistic or very limiting (Cossu et al., 2021). Recently, the idea of Class-Incremental with Repetition (CIR) scenarios has started to gather some attention in CL (Cossu et al., 2021). CIR scenarios are arguably more flexible in the definition of the stream, since they allow both the introduction of new classes and the repetition of previously seen classes. Crucially, repetition is a property of the environment and cannot be controlled by the CL agent. This is very different from Replay strategies (Hayes et al., 2021), where the repetition of previous concepts is heavily structured and can be tuned at will.

CIR defines a family of CL scenarios which ranges from CI (new classes only, without repetition) to DI (full repetition of all seen classes). Although appealing, currently there exists neither a quantitative analysis nor an empirical evaluation of CL strategies learning in CIR scenarios. Mainly, because it is not obvious how to build a stream with repetition, given the large amount of variables involved. How to manage repetition over time? How to decide what to repeat? What data should we use? In this paper, we provide two generators for CIR that, starting from a single dataset, allow to build customized streams by only setting few parameters. The generators are as easy to use as CI or DI ones.

We leveraged our generators to run an extensive empirical evaluation of the behavior of CL strategies in CIR scenarios. We found out that knowledge accumulation happens naturally in streams with

repetition. Even a naive fine-tuning, subjected to complete forgetting in CI scenarios, is able to accumulate knowledge for classes that are not always present in an experience. We observed that Replay strategies still provide an advantage in terms of final accuracy, even though they are not crucial to avoid catastrophic forgetting. On one side, distillation-based strategies like LwF (Li & Hoiem, 2018) are competitive in streams with a moderate amount of repetition. On the other side, existing Replay strategies are not specifically designed for CIR streams. We propose a novel Replay approach, called Frequency-Aware Replay (ER-FA) designed for streams with unbalanced repetition (few classes appear rarely, the other very frequently). ER-FA surpasses by a large margin other Replay variants when looking at infrequent classes and it does not lose performance in terms of frequent classes. This leads to a moderate gain in the final accuracy, with a much better robustness and a reduced variance across all classes. Our main contributions are:

1. The design of two CIR generators, able to create streams with repetition by only setting few control parameters. We built both generators with Avalanche (Lomonaco et al., 2021) and we will make them publicly available to foster future research. The generators are general enough to fit any classification dataset and are fully integrated with Avalanche pipeline to run CL experiments.

2. We perform an extensive evaluation of the properties of CIR streams and the performance of CL strategies. We study knowledge accumulation and we showed that Replay, although still effective, is not crucial for the mitigation of catastrophic forgetting. Some approaches (e.g., LwF) look more promising than others in CIR scenarios. We consolidate our results with an analysis of the CL models over time through Centered Kernel Alignment (CKA) (Kornblith et al., 2019) and weights analysis.

3. We propose a novel Replay variant, ER-FA, which is designed based on the properties of CIR scenarios. ER-FA surpasses other Replay strategies in unbalanced streams and provide a more robust performance on infrequent classes without losing accuracy on the frequent ones.

## 2    CLASS-INCREMENTAL LEARNING WITH REPETITION GENERATORS

CL requires a model to learn from a stream of $N$ experiences $\mathcal{S} = \{e_1, e_2, ..., e_N\}$, where each experience $e_i$ brings a dataset of examples $D_{e_i} = \{X_i, Y_i\}$. Many CL scenarios, like CI or DI, are generated from a fixed dataset $\mathcal{D} = \{(x, y); x \in X, y \in Y\}$, where $x$ is the input example, $y$ is the target and $Y = \{1, \cdots, C\}$ is the label space (closed-world assumption). Depending on how classes from the entire dataset $\mathcal{D}$ are shown or revisited in the stream, this configuration can lead to CI, CIR or DI scenarios (Figure 1). In Table 1, we formally present and compare the properties of the three scenario types.

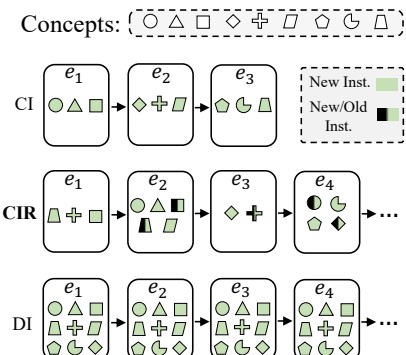

Figure 1: Illustration of scenario types that can be generated with episodic partial access to a finite set of concepts. The shape colors indicate whether instances are new in each episode or can be a mixture of old and new instances.

In CIR, streams with repetition are characterized by multiple occurrences of the same class over time. To study this scenario, we propose two stream generators designed to create a stream from a finite dataset: the Slot-Based Generator ($G_{slot}$) and the Sampling-Based Generator ($G_{samp}$). $G_{slot}$ generate streams by enforcing constraints on the number of occurrences of classes in the stream using only two parameters. $G_{slot}$ does not repeat already observed samples, therefore the stream length is limited by the number of classes. However, it guarantees that all samples in the dataset will be observed exactly once during the lifetime of the model. Instead, $G_{samp}$ generates streams according to several parametric distributions that control the stream properties. It can generate arbitrarily long streams in which old instances can also re-appear with some probability.

### 2.1    SLOT-BASED GENERATOR

The Slot-Based Generator $G_{slot}$ allows to carefully control the class repetitions in the generated stream with a single parameter $K$. $G_{slot}$ takes as input a dataset $\mathcal{D}$, the total number of experiences

| Property | CI | DI | CIR |
|---|---|---|---|
| Instance Repetition * | $X_{e_i} \cap X_j = \emptyset$ | $X_i \cap X_j = \emptyset$ | $|X_i \cap X_j| \geq 0$ |
| Domain Coverage | $\bigcup_{i=1}^{i=N} X_i = X$ | $\bigcup_{i=1}^{i=N} X_i = X$ | $\bigcup_{i=1}^{i=N} X_i \in \mathcal{P}(X) \setminus \emptyset$ |
| Concept Repetition * | $Y_i \cap Y_j = \emptyset$ | $Y_1 = \ldots = Y_N = Y$ | $|Y_i \cap Y_j| \geq 0$ |
| Codomain Coverage | $\bigcup_{i=1}^{i=N} Y_i = Y$ | $\bigcup_{i=1}^{i=N} Y_i = Y$ | $\bigcup_{i=1}^{i=N} Y_i \in \mathcal{P}(Y) \setminus \emptyset$ |

Table 1: Comparison of scenario properties in CI, DI and CIR. $\mathcal{P}(A)$ and $|A|$ represent the power set and the cardinality of set A. *: $\forall 1 \leq i, j \leq N$ , $i \neq j$.

$N$ and the number of slots per experience $K$. It returns a CIR stream composed by $N$ experiences, where each of the $K$ slots in each experience is filled with samples coming from a single class.

$G_{slot}$ constrains the slot-class association such that all the samples present in the dataset are seen exactly once in the stream. Therefore, $G_{slot}$ considers repetition at the level of concepts. To implement this logic, $G_{slot}$ first partitions all the samples in the dataset into the target number of slots. Then, it randomly assigns without replacement $K$ slots per experience. At the end, the $N \mod K$ blocks remaining are assigned to the first experience, such that the rest of the stream is not affected by a variable number of slots.

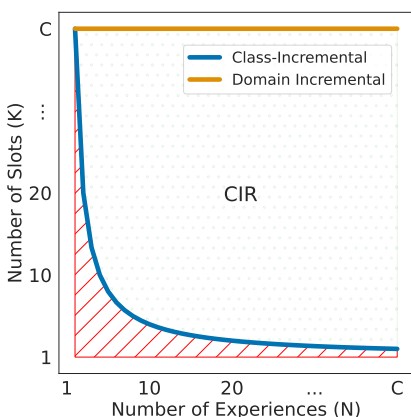

The Slot-Based Generator is useful to study the transition from CI scenarios to DI scenarios, obtained by simply changing the parameter $K$ (Figure 2). For example, let us consider a dataset with 10 classes such as MNIST. By choosing $N = 5$ and $K = 2$ we obtain the popular Split-MNIST, a CI scenario with no repetition and 2 classes for each experience. Conversely, by setting $N = 5$ and $K = 10$ we obtain a DI stream where all the 10 classes appear in each experience with new unseen samples. More in general, given a dataset with $C$ classes, we obtain a CI scenario by setting $K = \frac{C}{N}$ ($N$ must divide $C$). We obtain a DI scenario by setting $K = C$. In Appendix B we illustrate the overall steps of stream generation (Figure 12), and provide a step-by-step formal definition of $G_{slot}$ (Algorithm 2).

Figure 2: Illustration of how various scenarios can be generated by $G_{slot}$, by changing $K$ and $N$. The red area under the blue curve represents invalid scenarios.

## 2.2 SAMPLING-BASED GENERATOR

The Sampling-Based Generator ($G_{samp}$) generates arbitrarily long streams and controls the repetitions via probability distributions. The stream generator allows to control the *first occurrence* of new classes and the *amount of repetitions* of old classes. Unlike $G_{slot}$, it allows to generate infinite and even unbalanced streams.

$G_{samp}$ parameters:

- $N$: Stream length, i.e. number of experiences in the stream.
- $S$: Experience size which defines the number of samples in each experience.
- $\mathcal{P}_f(\mathcal{S})$: Probability distribution over the stream $\mathcal{S}$ used for sampling the experience ID of the first occurrence in each class.
- $P_r$: List of repetition probabilities for dataset classes.

Note that $\mathcal{P}_f$ is a probability mass function over the stream $\mathcal{S}$ which means it sums up to $1.0$ and determines in which parts of the stream it is more likely to observe new classes for the first time. However, the list of probabilities $\{p_1, p_2, ..., p_C\}$ in $P_r$ are independent and each probability value $0.0 \leq p_i \leq 1.0$ indicates how likely it is for each class to repeat after its first occurrence.

For each experience, $G_{samp}$ samples instances from the original dataset $\mathcal{D}$ according to a two step process. First, $G_{samp}$ defines a $C \times N$ binary matrix $T$ called *Occurrence Matrix* that determines which classes can appear in each experience. Then, for each experience $e_i, 1 \leq i \leq N$ we use the $i$-th column of $T$ to sample data points for that experience. The generator uses the inputs $N, \mathcal{P}_f(\mathcal{S})$

Figure 3: Schematic view of $G_{samp}$ generator. Each concept is shown with a different color.

and $P_r$ to generate $T$. Therefore, it first initializes $T$ as a zero $C \times N$ matrix. Then for each class $c$ in the dataset, it uses $\mathcal{P}_f(\mathcal{S})$ to sample the index of the experience in which class $c$ appears for the first time. Different probability distributions can be used to either spread the first occurrence along the stream or concentrate them at the beginning, which allows a good flexibility in the stream design. After sampling the first occurrences, the classes are then repeated based on $P_r$ probability values to finalize matrix $T$. In the simplest case, $P_r$ can be fixed to the same value for all classes to generate a balanced stream.

Once the matrix $T$ is constructed, a random sampler is used to sample patterns for each experience. Since each experience may contain an arbitrary number of classes, another control parameter that could be added here is the fraction of samples per class in experience size $S$. For simplicity we keep the fractions equally fixed and thus the number of datapoints sampled from each class in experience $e_i$ is $\lfloor \frac{S}{|\mathcal{C}^i|} \rfloor$ where $|\mathcal{C}^i|$ indicates the number of classes present in $e_i$. Since the sampler is stochastic, each time we sample from a class, both new and old patterns can appear for that class. Given a large enough stream length $N$, the final stream will cover the whole dataset with a measurable amount of average repetition. In Figure 3 we demonstrate the schematic of the generator $G_{samp}$. We provide the pseudo-code for $G_{samp}$ in Appendix C (Algorithm 2).

Although we assume a fixed number of instances per class in $\mathcal{D}$, $G_{samp}$ can be easily extended to settings where the number of instances can grow over time. Moreover, the sampler can also be designed arbitrarily depending on the stochasticity type, e.g., irregular or cyclic.

## 3 FREQUENCY-AWARE REPLAY

Experience Replay (ER) is the most popular CL strategy due to its simplicity of use and high performance in class-incremental scenarios. The storage policy, which determines which samples to keep in a limited buffer, is the major component of ER methods. Class-Balanced (CB) and Reservoir Sampling (RS) Vitter (1985) are the most popular storage policies in ER methods. CB keeps a fixed quota for each class, while RS samples randomly from the stream, which leads to the class frequency in the buffer being equal to the frequency in the stream. CB and RS

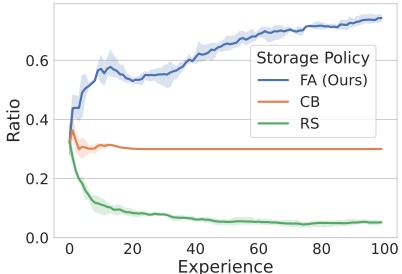

Figure 4: Ratio of buffer slots for infrequent classes for three random seeds.

are great choices for balanced streams such as class incremental scenarios, where the number of samples per class is the same over the whole stream. However, as in most real-world scenarios, CIR scenarios are naturally unbalanced, and different classes may have completely different repetition frequencies. Accordingly, CB and RS storage policies may suffer a big accuracy drop in the infrequent classes of an unbalanced stream. For example, in highly unbalanced streams, RS will store an unbalanced buffer replicating the the original distribution of the stream, which is sub-optimal because the less frequent classes will require more repetition to prevent forgetting, while the frequent classes will be repeated naturally via the stream occurrences.

We propose *Frequency-Aware* (FA) storage policy that addresses the imbalance issue in CIR streams by online adjustment of the buffer slots in accordance with the amount of repetition for each class. Given a buffer $B$ with a maximum size of $M$, a list of previously observed classes $P$ initialized as $P = \{\}$ with a corresponding list $O$ indicating the number of observations per class $c$ in $C$, and a dataset $D_i$ from experience $e_i$, the algorithm first checks the present classes $P_i$ in $D_i$ and adds them to $P$ ($P \leftarrow P \cup P_i$). Then, for each class $c$ in $P_i$ it increments the number of observations $O[c]$ by

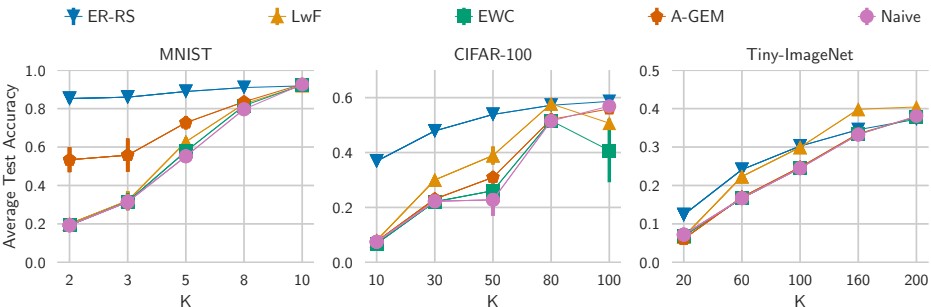

Figure 5: Average Test Accuracy for different values of $K$ in CIR scenarios generated with $G_{slot}$. Class-Incremental scenarios are represented by the left-most point of each plot, Domain-Incremental scenarios by the right-most point. Results averaged over 3 seeds.

1 if the class was previously seen, otherwise it initializes $O[c] = 1$. After updating the number of observations, FA computes the buffer quota $Q$ for all observed classes by inverting the number of observations ($Q = \lceil \frac{1}{O[c]} \forall c \in C \rceil$) and normalizes it . This way, the algorithm offers the less frequent classes a larger quota. Finally, a procedure ensures the buffer is used to its maximum capacity by filling unused slots with samples from more frequent classes sorted by their observation times. This is a crucial step since it is possible that an infrequent class which is not present $e_i$ will be assigned with a larger quota than its current number of samples in $B$, and therefore the buffer will remain partially empty. In Figure 4 we show how our method assigns higher ratio of samples for infrequent classes to overcome the imbalance issue in the stream. For further analysis and pseudo-code of FA policy refer to Appendix E. We present examples of unbalanced scenarios in Appendix D.

## 4 EMPIRICAL EVALUATION

We study CIR scenarios by leveraging our generators $G_{samp}$ and $G_{slot}$. First, by using $G_{slot}$ we provide quantitative results about forgetting in CL strategies when transitioning from CI to DI scenarios (Sec. 4.1). Then, by using $G_{samp}$ we focus on long streams with 500 experiences and study the performance of Replay and Naive (Sec. 4.2). The long streams give us the opportunity to study knowledge accumulation over time in the presence of repetition. We also provide an intuitive interpretation of the model dynamics over long streams (Sec. 4.3). Finally, we show that our Frequency-Aware Replay is able to exploit the repetitions present in the stream and to surpass the performance of other replay approaches not specifically designed for CIR scenarios (Sec. 4.4).

The experiments were conducted using the CIFAR-100 Krizhevsky et al. (2009) and Tiny-ImageNet LeCun et al. (1998) datasets with the ResNet-18 model. For $G_{slot}$, we run experiments for Naive (incremental fine tuning), LwF Li & Hoiem (2018), EWC Kirkpatrick et al. (2017), Experience Replay with reservoir sampling Kim et al. (2020) (ER-RS) and AGEM Chaudhry et al. (2018) strategies. For $G_{samp}$ we run experiments for Naive and ER (CB/RS/FA) strategies. We set the default buffer size for CIFAR-100 to 2000, and for Tiny-ImageNet to 4000 in the replay strategies. We evaluate all strategies on the Average Test Accuracy (TA).

### 4.1 TRANSITION FROM CLASS-INCREMENTAL TO DOMAIN INCREMENTAL

DI and CI scenarios are heavily studied in the CL literature. However, little is known about what happens to the performance of popular CL strategies when gradually transitioning from one scenario to the other. By changing the value of $K$ in $G_{slot}$, we provide a quantitative analysis of such behavior in CIR scenarios. Figure 5 shows the Average Test Accuracy over all classes for different CL strategies when transitioning from CI (left-most point of each plot) to DI (right-most point of each plot).

Replay is one of the most effective strategies in CI scenarios. As expected, in CIR scenarios the advantage provided by ER-RS with respect to other CL strategies diminishes as the amount of repetition increases. However, in order for the other strategies to match the performance of ER-RS, the environment needs to provide a large amount of repetition.

LwF guarantees a consistent boost in the performance, both in CIFAR-100 and Tiny-ImageNet. In particular, and quite surprisingly, on Tiny-ImageNet LwF is able to quickly close the gap with ER-RS and even surpass it as the amount of repetition increases. The positive interplay between

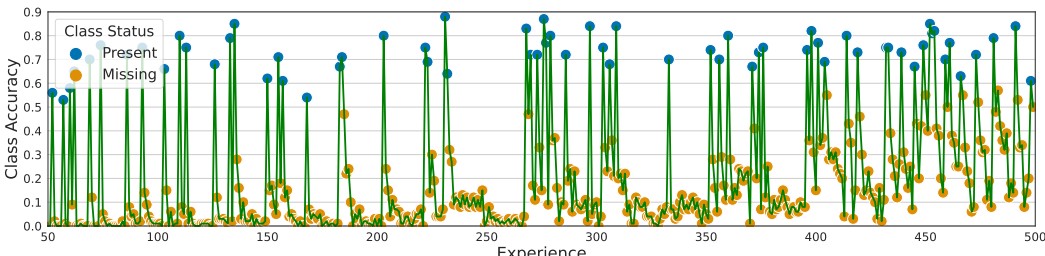

Figure 6: Accuracy of a particular class over the stream. The target class is either present or absent in the experiences indicated by the blue and orange points, respectively.

distillation and repetition provides an effective way to mitigate forgetting in CIR scenarios, without the need to explicitly store previous samples in an external memory. EWC showed different sensitivity to the regularization hyper-parameter $\lambda$. We experimented with $\lambda = 0.1, 1, 10, 100$. While on MNIST we did not see any difference in performance, on CIFAR-100 and Tiny-ImageNet large values of $\lambda$ lead to a dramatic decrease, dropping as low as Naive. We found $0.1$ to be the best value on both CIFAR-100 and Tiny-ImageNet. This configuration only provides a low amount of regularization. Overall, the role played by the natural repetition already guarantees a sufficient stability of the model, which is additionally boosted only in the case of LwF.

## 4.2 IMPACT OF REPETITION IN LONG STREAMS

We investigate the impact of repetition in long streams ($N = 500$) generated with $G_{samp}$. For the long-stream experiments we also report the missing-classes accuracy (MCA) and seen-classes accuracy (SCA). MCA measure the accuracy over the classes that were seen before but are missing in the current experience, and SCA measure the accuracy over all seen classes up to the current experience.

**Missing Class Accuracy Increases Over Time** In CI scenarios, a Naive strategy catastrophically forgets each class as soon as it starts learning on new classes. Surprisingly, we found that in CIR scenarios there is knowledge accumulation over time for all the classes. Figure 6 shows the accuracy of a single class over time, highlighting whether the class is present or not in the current experience. At the beginning of the stream missing classes are completely forgotten, which can be noticed by the instant drop of the accuracy to zero. However, over time the model accumulate knowledge and the training process stabilizes. As a result, the accuracy of missing classes tends to increase over time, suggesting that the network becomes more resistant to forgetting. Notice that this is an example of continual learning property that is completely ignored when testing on CI scenarios. This finding prompts the question, *"What is happening to allow knowledge accumulation even for Naive finetuning?"*. We investigate this question by analysing the model's accuracy over time and the properties of the learned model in the next experiments.

**Accuracy Gap Between Naive and Replay Decreases Over Time** To study the impact of long streams with repetitions we monitor the accuracy gap between ER and Naive fine-tuning by comparing their accuracy after each experience. For the scenario configuration, we set $\mathcal{P}_f(\mathcal{S})$ as a *Geometric* distribution with a coefficient of $0.01$ and fix the probability of repetition $\mathcal{P}_r$ as $0.2$ for all classes. For more details and illustrations on distribution types refer to Appendix C. In such scenarios, the majority of classes occur for the first time in the first quarter of the stream, and then repeat with a constant probability of $0.2$ which makes them appropriate for our experiments since all classes are observed before the middle of the stream and the repetition probability is low enough.

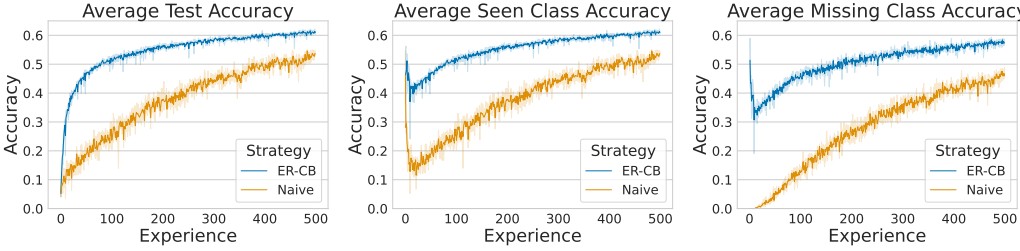

Figure 7: Average test accuracy and average missing class accuracy plots for long streams streams with 500 experiences.

As can be seen in Figure 7, while the accuracy of ER saturates over time, the accuracy of Naive increases, closing the gap between the two strategies from around $25\%$ in experience 100 to $7\%$ in experience 500. This supports our hypothesis that neural network's ability to consolidate knowledge is significantly influenced by "natural" repetition in the environment.

**The Role of Repetition**   The amount of repetition is one of the key aspects of a CIR scenario. To find out how strategies perform under different repetition probabilities, we consider a setting where all components of a scenario are fixed except for $P_r$. For this experiment, we set $\mathcal{P}_f(\mathcal{S})$ as geometric distribution with $p = 0.2$ and let $P_r$ change. In Figure 8 we demonstrate the seen class accuracy (SCA) for the Naive and ER-CB strategies in CIFAR-100. It is clear from the plots, that the model's rate of convergence can be significantly affected by the amount of repetition in the scenario. Although, it may seem obvious that higher repetition leads to less forgetting, it is not very intuitive *to what extent* different strategies may gain from the environment's repetition. While the Naive strategy gains a lot from increased repetition, the replay strategy saturates after some point for higher repetitions and the gaps close between different repetition values.

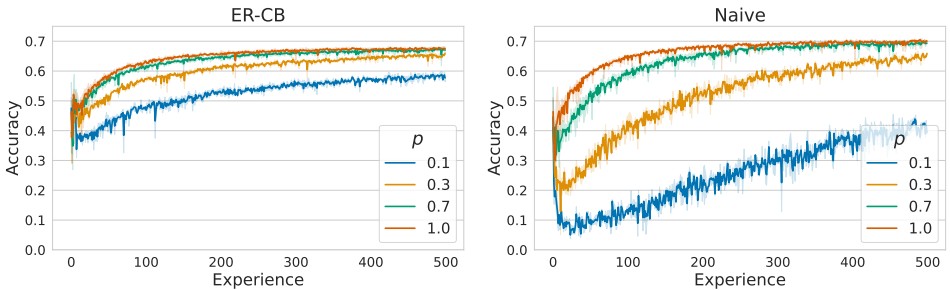

Figure 8: Retained accuracy for different values of $p$ in $P_r$.

## 4.3   MODEL SIMILARITY AND WEIGHT SPACE ANALYSIS

**Weight Interpolation**   Based on the "gradual loss drop" observation in missing classes, we study how the loss surface changes over time if we perturb the weights. We interpolate between the model weights from two consecutive checkpoints with an interval of 10 experiences in various segments of the stream. Assuming that $w_t^*$ and $w_{t+10}^*$ are the obtained solutions for experiences $t$ and $t + 10$ respectively, we generate eight in-between models $w_k = \alpha * w_t^* + (1 - \alpha) * w_{t+50}^*$ by by increasing $\alpha$ from zero to one, and then compute the accuracy of $w_k$ for the data of experience $t$. We show the interpolation accuracy for various pairs of experiments in different segments of the stream for the Naive strategy in Figure 9 (left). In the beginning of the stream, the accuracy of experience $t$ in each pair drops significantly, while we observe a milder loss drop towards the end of the stream. The findings suggest that, towards the end of the stream, even a relatively big perturbation does not have a large negative effect on the model's accuracy and the optimal solutions of the consecutive experiments are connected with a linear low-loss path.

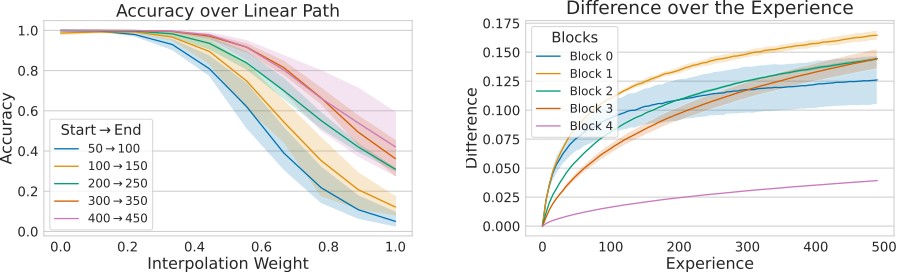

Figure 9: (left) Interpolation accuracy. (right) Weight changes in each block. The difference used in (right) is calculated as $D_j = \frac{1}{|\theta_0|} \sum_i^{\theta_b} \left\| \frac{(\theta_{0,i} - \theta_{j,i})}{\|\theta_{0,i}\|_2} \right\|$, where the weights of experience $j$ are compare with the initialization $\theta_0$ for each block $i$

**Weight changes**   Another approach to analyzing the gradual drop of the accuracy is by dissecting how much, when, and where the weight changes occurs. As shown in Figure 9 (right), we can observe that within the first experiences, there is a significant difference for blocks 0, 1, and 2. This

| DS | Strategy | Fraction= 0.1 | | Fraction= 0.3 | | Fraction= 0.5 | |
| --- | --- | --- | --- | --- | --- | --- | --- |
| | | MCA | TA | MCA | TA | MCA | TA |
| C-100 | Naive | $5.0 \pm 0.7$ | $58.0 \pm 0.1$ | $7.3 \pm 2.2$ | $49.0 \pm 0.8$ | $8.0 \pm 2.0$ | $40.8 \pm 1.4$ |
| | ER-RS | $11.4 \pm 0.9$ | $57.7 \pm 0.7$ | $16.7 \pm 3.6$ | $51.1 \pm 0.4$ | $20.5 \pm 1.9$ | $45.6 \pm 0.8$ |
| | ER-CB | $30.9 \pm 2.7$ | $59.5 \pm 0.1$ | $34.5 \pm 1.7$ | $55.3 \pm 0.1$ | $35.7 \pm 0.5$ | $52.0 \pm 1.5$ |
| | ER-FA | $\mathbf{52.2 \pm 1.1}$ | $\mathbf{60.8 \pm 0.3}$ | $\mathbf{44.7 \pm 1.5}$ | $\mathbf{57.8 \pm 0.4}$ | $\mathbf{40.9 \pm 1.2}$ | $\mathbf{54.2 \pm 1.2}$ |
| TIN | Naive | $2.0 \pm 0.8$ | $\mathbf{33.5 \pm 0.4}$ | $2.0 \pm 0.1$ | $29.1 \pm 0.1$ | $2.0 \pm 0.1$ | $24.0 \pm 0.4$ |
| | ER-RS | $3.7 \pm 0.6$ | $31.8 \pm 1.2$ | $4.4 \pm 0.7$ | $28.1 \pm 0.2$ | $6.0 \pm 0.1$ | $24.0 \pm 0.1$ |
| | ER-CB | $10.4 \pm 0.2$ | $32.2 \pm 0.7$ | $10.0 \pm 1.0$ | $28.8 \pm 0.2$ | $11.0 \pm 0.2$ | $26.0 \pm 0.3$ |
| | ER-FA | $\mathbf{22.0 \pm 1.0}$ | $33.0 \pm 0.9$ | $\mathbf{15.3 \pm 1.0}$ | $\mathbf{30.4 \pm 0.1}$ | $\mathbf{13.6 \pm 0.1}$ | $\mathbf{27.0 \pm 0.1}$ |

Table 2: Unbalanced scenario results for the CIFAR-100 (C-100) and TinyImageNet (TIN) dataset. "Fraction" refers to the fraction of infrequent classes having repetition probability of only 10%.

difference then stalls, showing that as we continue training experiences, the weights of these blocks stabilize. On the other hand, blocks 3 and 4 show a linear increase in the difference with the number of experiences. An explanation for this phenomenon, is that the first layers of the model capture knowledge that can be useful for several classes (more general), so it is unnecessary to change them after several experiences. On the other hand, the last blocks are the ones that memorize or learn more specific patterns, so they adapt to each experience.

**CKA Analysis** Finally, we show the CKA Kornblith et al. (2019) of the model in the beginning, middle and the end of the stream with an interval difference of 50 experiences. As shown in the visualizations in Figure 10, the longer the model is trained on more experiences, the less significant the changes in the representations become especially for the final layers. We can see that the diagonal of the CKA becomes sharper propagating forward with more experiments. This indicates that although the model is trained on different subsets of classes in each experiment, the representations change less after some point in the stream.

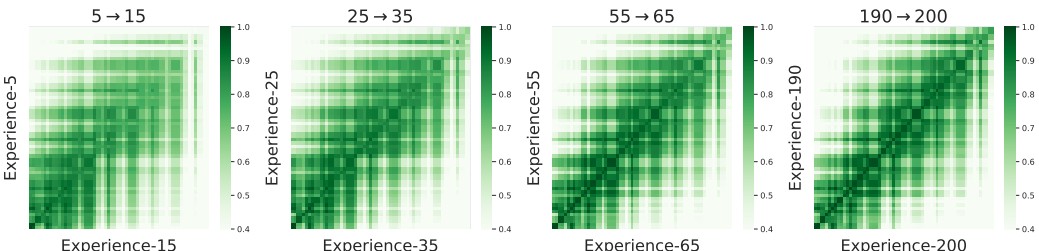

Figure 10: CKA of the model in different parts of the stream.

## 4.4 FREQUENCY-AWARE REPLAY IN UNBALANCED SCENARIOS

We conduct experiments for bi-modal unbalanced scenarios where classes can have a high frequency of $1.0$ or a low frequency of $0.1$. We use a fraction factor that determines the amount of infrequent classes in the scenario, e.g., Fraction=0.3 means that $30\%$ of the classes are infrequent. In Table 2 we compare ER-FA with the Naive, ER-RS and ER-CB strategies. The numbers show the MCA and average Test Accuracy (TA) metrics for each strategy in the end of the stream averaged over three runs. Our strategy outperforms all other scenarios in almost all settings in both CIFAR-100 and TinyImageNet datasets in terms of TA, and significantly outperforms other methods in terms of MCA (in the last experience). Moreover, we

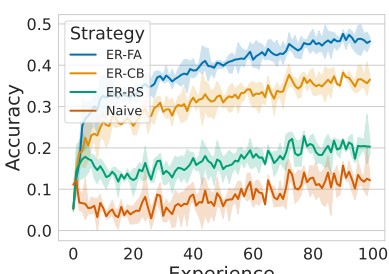

Figure 11: Accuracy of Infrequent Classes.

illustrate the accuracy of infrequent classes in CIFAR-100 experiments for Fraction=0.3 in Figure 11 where ER-FA achieves considerably higher accuracy in the whole stream by assigning larger quota to infrequent classes without losing its performance on frequent classes (refer to Appendix G for further illustrations).

## 5 RELATED WORK

Current CL methods are mainly focused on two types of benchmarks namely, *Multi Task* (MT) and *Single Incremental Task* (SIT) Maltoni & Lomonaco (2019). MT divides training data into distinct tasks and labels them during training and inference. SIT splits a single task into a sequence of unlabeled experiences. SIT can be further divided into Domain-Incremental (DI) where all classes are seen in each experience, and Class-Incremental (CL) where each experience contains only new (unseen) classes van de Ven & Tolias (2019). Both DI and CI are extreme cases and are unlikely to hold in real-world environments Cossu et al. (2021). In a more realistic setting, the role of natural repetition in CL scenarios was studied in the context of New Instances and Classes (NIC) scenario Lomonaco et al. (2020) and the CRIB benchmark Stojanov et al. (2019). NIC mainly focuses on small experiences composed of images of the same object, and repetitions in CRIB are adapted to a certain dataset and protocol. The Class-Incremental with Repetition (CIR) scenario was initially formalized in Cossu et al. (2021), however the work lacks a systematic study of CIR scenarios as the wide range of CIR scenarios makes them difficult to study.

To counter the lack of repetition in CI, replay has been extensively used as a CL strategy (Rebuffi et al., 2017; Lopez-Paz & Ranzato, 2017; Chaudhry et al., 2018; Wu et al., 2019; Castro et al., 2018; Belouadah & Popescu, 2019; Kim et al., 2020; Douillard et al., 2020). In such methods, natural repetition is artificially simulated by storing past data in an external memory, and replaying them alongside the scenario stream data. Repetition reduces catastrophic forgetting through implicit regularization of model's weights Hayes et al. (2021). In CI benchmarks, replay seems to be the only working strategy van de Ven et al. (2020). In other words, replay seems to be a necessity when no natural repetition happens.

Although replay can be seen as a method to simulate natural repetitions artificially, the two concepts are fundamentally different. Repetition in replay strategies occurs with the same data seen in previous experiences, which is neither realistic nor biologically plausible Gupta et al. (2010). On the other hand, natural repetitions of already seen objects occur in different real-world environments, and better fit the CIR scenario studied in this paper. Recently, Lesort et al. (2022) scaled the number of tasks in a finite world setting (Boult et al., 2019; Mundt et al., 2022) where the model has access to a random subset of classes in each experience. The authors proposed naive fine-tuning with masking techniques to improve retained accuracy. Our work is different in the sense that we compare among different strategies and study various types of repetitions with two flexible generators.

## 6 DISCUSSION AND CONCLUSION

We defined CIR scenarios which represent CL environments where repetition is naturally present in the stream. Although the concept of repetition is quite intuitive, it is not obvious how to realize it in practice for research purposes. Therefore, we proposed two CIR generators that can be exploited to address this issue. Through empirical evaluations, we showed that, unlike CI scenarios, knowledge accumulation happens naturally in CIR streams, even without applying any CL strategy. This raised the question of whether the systematic repetition provided by Replay is critical in all CIR scenarios. With several experiments in long streams, we demonstrated that although Replay provides an advantage in general, even random repetition in the environment can be sufficient to induce knowledge accumulation given a long enough lifetime.

Moreover, we found that existing Replay strategies are exclusively designed for classical CI scenarios. Thus, we proposed a novel strategy, ER-FA, to exploit the properties of CIR scenarios. ER-FA accumulates knowledge even in highly unbalanced stream in terms of class frequency. ER-FA outperforms by a large margin other Replay approaches when monitoring the accuracy for infrequent classes while preserving accuracy for the frequent ones. Overall, ER-FA guarantees a more robust performance on a wide range of real-world scenarios where classes are not homogeneously distributed over time.

The framework defined in this work opens new research directions which depart from the existing ones, mainly focused on the mitigation of forgetting in CI scenarios. We hope that our experiments and results will promote the study of CIR scenarios and the development of new CL strategies, able to exploit the inner semantics of repetition, a natural trait of real-world data streams, in many other clever and imaginative ways.

## 7 REPRODUCIBILITY STATEMENT

We will provide an open-source implementation of our generators and algorithms with the scripts needed to reproduce the results reported in the paper.

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

## A    RELATED WORK (CONTINUED)

Considering benchmark formalization frameworks, De Lange & Tuytelaars (2021) recently proposed a subdivision aimed at framing continual learning setups by categorizing them based on the batch and observable horizon that the learning agent is able to access at each time. With this framework, the authors aim to better formalize the online learning setup. While the concept of observable horizon may be useful in evaluating the significance and local (in time) usefulness of natural repetition in a training stream, this work does not consider the concept of natural repetition in its framework.

Recently, Koh et al. (2022) proposed to introduce blurry task boundaries in class incremental benchmarks. Their proposal is based on previous works Bang et al. (2021); Aljundi et al. (2019) that tried to produce more realistic benchmarks by blurring the class-incremental scenario, which however resulted in a setup in which no classes are added to new tasks. They argue that this idea moves the focus too far away from the class-incremental setup and it is still not quite realistic. The resulting setup, named i-Blurry, aims at resolving the aforementioned issues and moving toward a more realistic scenario by partitioning the classes available in the source dataset into two groups: Disjoint and Blurry. Classes of the disjoint group are gradually added in successive experiences while samples of classes from the blurry group always appear in all experiences with their numerosity being controlled through a blur ratio $M$. The authors show that, based on the degree of disjunction $N$ and blurry $M$, this framework can produce class-incremental (no blurring), domain-incremental (no disjunction), and blurred setups. This setup is the one that most moved towards the direction of introducing repetition in Continual learning benchmarks in a controlled way so far. However, the proposed blurring mechanism is too coarse-grained to simulate a natural repetition of concepts as the significance of the repetition introduced by blurring relies too much on i) a random (uniform) sampling of the concepts to be repeated, ii) the static subdivision of classes in the Disjoint and Blurry groups.

## B    SLOT-BASED GENERATOR

Following the properties of CIR scenarios in Section 2, $G_{slot}$ generates a subset of CIR streams that hold the assumptions below in the defined properties:

- $|X_i \cap X_j| = 0$ : new instances appear in each experiences
- $\bigcup_{i=1}^{i=N} X_i = X$: all samples are used
- $|Y_i \cap Y_j| \geq 0, \forall 1 \leq i, j \leq N$ where $i \neq j$
- $\bigcup_{i=1}^{i=N} Y_i = Y$: all classes are used
- $X$ is constant.

These assumptions allow transitioning through different CIR scenario types between the two extremes of CI and DI.

### B.1    ALGORITHM

The overall steps of $G_{slot}$ are illustrated in Figure 12. In Algorithm 2 we present all steps of $G_{slot}$ used to generate arbitrary CIR scenarios given a dataset $D$, number of experiences $N$ and number of slots $K$. The output of the algorithm is a CL stream.

### B.2    TRANSITIONING

Transitioning in $G_{slot}$ for a scenario with a fixed number of experiences can be done by increasing $K$. When $K = 1$ the generated scenario will be class-incremental and as $K$ get closer to the total number of classes in $D$, the scenarios moves towards a domain incremental setting. In Figure 13 we show an example of how generated scenarios change by increasing $K$.

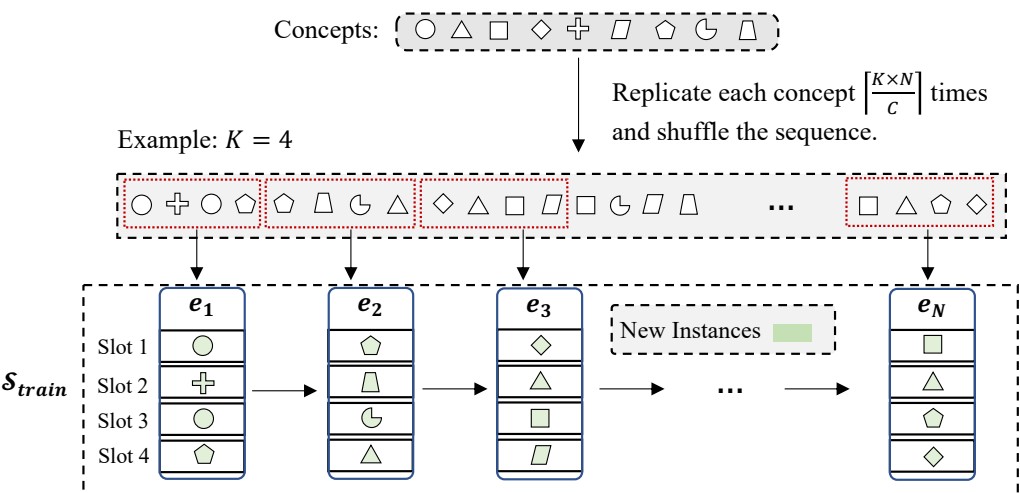

Figure 12: Illustrations of the overall steps of $G_{slot}$. Each shape represents a concept, and the green color means that new instances of that concept are used in each experience.

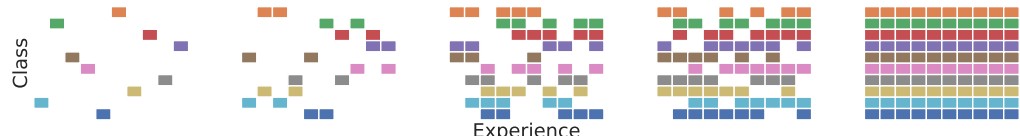

Figure 13: From left to right: transitioning from CI to DI in $G_{slot}$. Each class is represented with a unique color.

## C  SAMPLING-BASED GENERATOR

Following the properties of CIR scenarios in Section 2, $G_{samp}$ generates a subset of CIR scenarios that hold all defined properties. Additionally, for any stream $\mathcal{S} = \{e_1, e_2, ..., e_N\}$, $G_{samp}$ defines a probability distribution for the first occurrence of concepts over $S$ and per-class probabilities for each concept $c \in Y$. $G_{samp}$ can generate arbitrarily long stream ($N \geq 1$) and even from a growing set of samples $X$ where $Y$ remains constant.

### C.1  ALGORITHM

The overall steps of the $G_{slot}$ are shows in Algorithm 2.

### C.2  DISTRIBUTION TYPES

In this section we show some examples of different discrete distributions that can be used for $\mathcal{P}_f(\mathcal{S})$ and $P_r$. For $P_r$ we use the unnormalized version of the final distribution. Distributions used for $G_{samp}$ can be any arbitrary discrete distribution and are not limited to the ones we describe here.

#### C.2.1  ZIPFIAN

Given the number of elements $N$ and scalar $e \geq 0$ , the probability mass function of a Zipfian distribution over a list of $N$ elements is defined in Equation 1. When used for the probability of first occurrence, the distribution can be defined over the experiences of a stream. For example, $N$ can be considered as the number of experiences and $i$ can indicate the $i$th experience in the stream. By increasing $e$, the distribution over the stream will be skewed towards the beginning. In figure 14 we demonstrate some examples of first occurrence probabilities over a stream of length $10$ generated with Zipf distribution with increasing values of $e$. Many natural distribution follow Zipf distribution

---

**Algorithm 1** Slot-Based Generator ($G_{slot}$) Pseudo-Code.

---

**Require:** Dataset $D = \{(x_i, y_i)\}_{i=1,...,P}$ with $C$ classes, number of experiences $N$, experience size $S$ present in each experience.
**Ensure:** $K \leq N$
**Ensure:** $C \mod N = 0$
**Ensure:** $NK \mod C = 0$
  cls-idxs = {}                                         ▷ Empty dictionary
  **for** $y \in set(\{y_i\}_{i=1,...,P})$ **do**
      cls-idxs[$y$] = []                                 ▷ Empty list init
  **end for**
  **for** $i = 1, \ldots, P$ **do**
      cls-idxs[$y_i$].append($i$)
  **end for**
  slots={}                                             ▷ Empty dictionary
  **for** $y \in$ cls-idxs **do**
      slots[$y$] = []
      ksample = int(len(cls-idxs[$y$]) $/K$)
      **for** $k = 1, \ldots, \frac{N \times K}{C}$ **do**
         subset-idxs = pop(cls-idxs[$y$], ksample)
         subset-samples = [$x_{idx}$ for idx $\in$ subset-idxs]
         slots[$y$].append(subset-samples)
      **end for**
  **end for**
  stream = []
  **for** $n = 1, \ldots, N$ **do**
      experience = dataset()
      seen-classes = []
      **for** k=1,\ldots,K **do**
         **repeat**
             $y$ = sample(slots)
         **until** $y \notin$ seen-classes
         seen-classes.append($y$)
         experience.add(pop(slots[$y$], 1))
      **end for**
      stream.append(experience)
  **end for**
    **return** stream

---

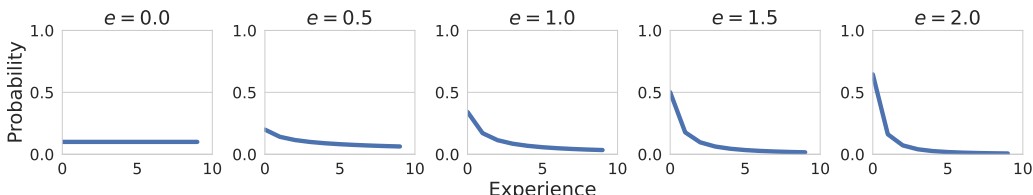

Figure 14: Zipf distribution with varying values of $e$.

and it can be used to generate highly skewed distributions both for first occurrence and repetition probabilities.

$$f(i; e, N) = \frac{\frac{1}{i^e}}{\sum_{n=1}^{N} \frac{1}{n^e}} \quad (1)$$

---

**Algorithm 2** Sampling-Based Generator ($G_{samp}$) Pseudo-Code.

---

**Require:** Dataset $D = \{(x_i, y_i)\}_{i=1,\dots,P}$ with $C$ classes, number of experiences $N$, number of slots $K$, probability distribution for first occurrence $\mathcal{P}_f(\mathcal{S})$, and list of repetition probabilities $P_r$.

$T = \{0\}_{C \times N}$         ▷ Initialize occurrence matrix with zeros
  **for** $c \in \{0, 1, \dots C\}$ **do**
    $i \sim \mathcal{P}_f(\mathcal{S})$         ▷ Samples the first occurrence of class $c$
    $T[c, i] = 1$
    **for** $j \in \{i, i+1, \dots N\}$ **do**
      $r \sim U(0, 1)$         ▷ $U$: uniform distribution over $[0, 1.0]$
      **if** $r < P_r[c]$ **then** $T[c, j] = 1$
      **end if**
    **end for**
  **end for**
    $E = \{\}$
  **for** $e_i \in \{1, 2, \dots N\}$ **do**
    $C_i = RetrieveClasses(e_i)$
    $D_{e_i} = Sample(D, C_i, S)$         ▷ Sample $S$ instances from dataset $D$ for classes $C_i$
    $E \leftarrow E \cup D_{e_i}$
  **end for**
  $\mathcal{S}_{train}, \mathcal{S}_{test} = GenerateStream(E)$         ▷ Generate streams using E
    **return** $\mathcal{S}_{train}, \mathcal{S}_{test}$

---

### C.2.2 POISSON

The PMF for Poisson distribution is given in Equation 2 where $\mu \geq 0$. Poisson with larger values of $\mu$ can be used for distributions where the probability of occurrence/repetition first rises and then gradually decreases over time.

$$f(i; \mu) = \frac{\mu^i e^{-\mu}}{i!} \tag{2}$$

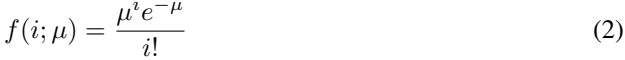
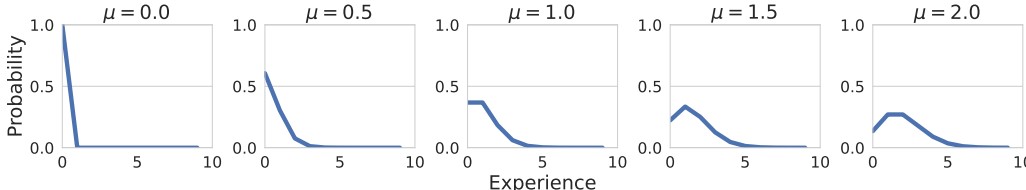

Figure 15: Poisson distribution with varying values of $\mu$.

### C.2.3 GEOMETRIC

Another useful distribution that can be used for the first occurrence probabilities over a stream is Geometric distribution with its PMF given in Equation 3. This distribution is in particular interesting for transitioning from domain incremental to class incremental. By setting $p = 1$, only the probability of experience $i = 0$ will be equal to $1.0$ and the rest will be zero, and by decreasing $p$, the probability will spread over the stream. In figure 17 we show examples for generated scenarios with $G_{samp}$ with Geometric first occurrence and fixed probability of repetition.

$$f(i, p) = (1 - p)^{i-1} p \tag{3}$$

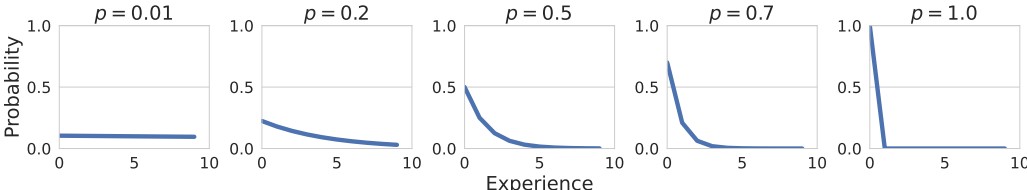

Figure 16: Geometric distribution with varying values of $p$.

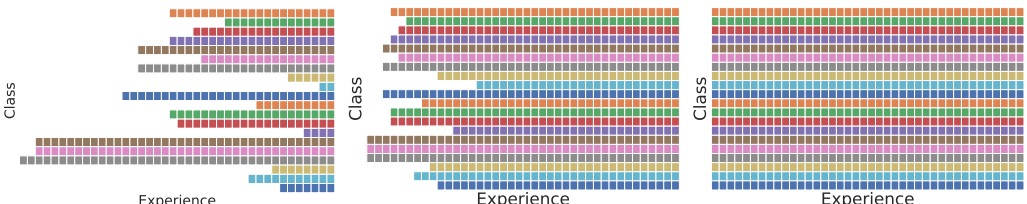

Figure 17: Scenarios generated with Geometric first occurrence and probability of repetition equal to $1.0$ for all classes. The $p$ values for the Geometric distributions from left to right are $0.01$, $0.2$ and $1.0$ respectively.

## D    UNBALANCED SCENARIOS

In this section we present a particular type of unbalanced scenarios where a subset of classes in the stream have a low probability of repetition and the rest repeat very often. We refer to such scenarios bi-modal scenarios, where each mode refers to a subset of classes with a distinct repetition probability. More specifically, we have a stream of experiences $\mathcal{S} = \{e_1, e_2, ..., e_N\}$ where $Y^S = \bigcup_1^N Y_{e_i}$ indicates the set of all available concepts in $\mathcal{S}$. In bi-modal scenarios $Y = Y^{if} \cup Y^{fr}$ where $Y^{if}$ and $Y^{fr}$ are the set of frequent and infrequent concepts respectively, and $Y^{if} \cup Y^{fr} = \emptyset$. In Figure 18 we show examples of unbalanced bi-modal scenarios.

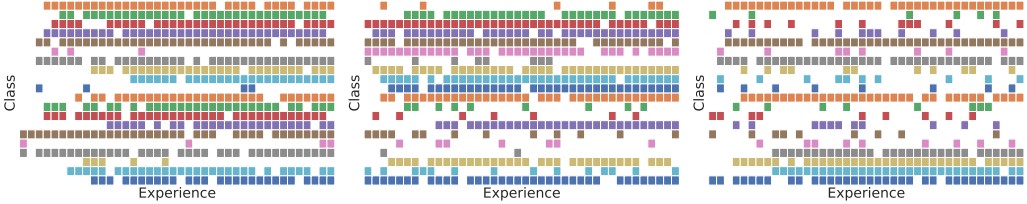

Figure 18: Unbalanced scenarios with two modes of repetition. The fractions of infrequent classes from left to right are $0.2$, $0.4$ and $0.6$ respectively. Repetition probabilities for frequent and infrequent classes are set to $0.2$ and $0.9$ accordingly.

## E    FREQUENCY-AWARE REPLAY

### E.1    ALGORITHM

In Algorithm 3 we present the steps for updating the buffer in FA storage policy.

### E.2    ANALYSIS: VARYING THE FRACTION OF INFREQUENT CLASSES

In this section, we study the behavior of FA, CB, and RS storage policies by changing the fraction of infrequent classes. In our analysis, we consider an unbalanced stream generated with $G_{samp}$ where $N = 100$ and the probability of repetition for frequent and infrequent classes are $0.9$ and $0.1$, respectively. In such streams, the large probability gap between frequent and infrequent classes

---

**Algorithm 3** Frequency-Aware Buffer.

---

**Require:** Current Buffer Set $B$, Maximum buffer size $M$, List of Seen Classes $C$, Number of Observation per Seen Class $O$
$\quad D = \text{GetExperienceDataset}(e_i)$
$\quad P = \text{DetectPresentClasses}(D)$
$\quad C \leftarrow C \cup P$
$\quad$**for** $c \in P$ **do** $\qquad\qquad\qquad\qquad$ ▷ For each present class, increment the number of observations
$\qquad$**if** $c \in O$ **then** $O[c]+ = 1$
$\qquad$**else** $O[c] = 1$
$\qquad$**end if**
$\quad$**end for**
$\quad Q = [\frac{1}{O[c]} \forall c \in C]$ $\qquad\qquad\qquad\qquad\qquad\qquad\qquad$ ▷ Calculate quota per class
$\quad \hat{Q} = \frac{Q}{|Q|}$ $\qquad\qquad\qquad\qquad\qquad\qquad\qquad\qquad$ ▷ Normalize quota values
$\quad S = \{\lceil Q[c] * M\rceil \forall c \in C\}$ $\qquad\qquad\qquad$ ▷ Calculate buffer slot size for each class
$\quad \text{UpdateSlots}(S)$ $\qquad\qquad\qquad$ ▷ Update assigned slots according to the current state of $B$
$\quad \text{UpdateBuffer}(B, D, S)$
$\qquad$**return** $B, M, C, O$

---

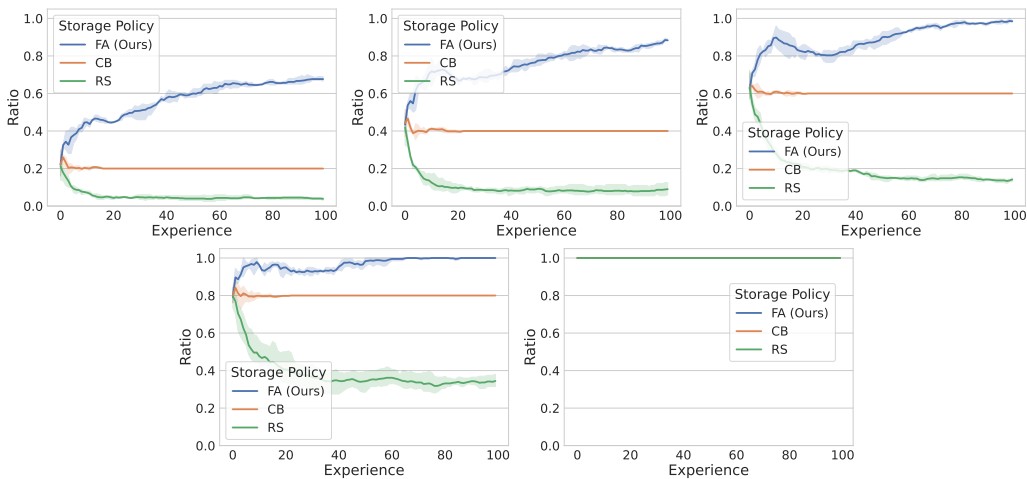

Figure 19: Ratio of samples for infrequent classes in unbalanced scenarios for the FA, CB and RS policies. Fraction of infrequent classes from top-left to bottom-right are $20\%, 40\%, 60\%, 80\%, 100\%$.

helps us observe the difference more clearly. We report the ratio of samples assigned to infrequent classes in the buffer in the lifetime of the model in the stream for scenarios where the fraction of infrequent classes is equal to $\{20\%, 40\%, 60\%, 80\%, 100\%\}$. For this experiment, we set the buffer size to $500$ for all methods.

As demonstrated in Figure 19, when the fraction of infrequent classes is equal to $20\%$, i.e. only $20\%$ of classes are infrequent, the ratio is very low for RS policy as it tries to replicate the true distribution of the stream while CB assigns exactly $20\%$ of the buffer space to the infrequent samples. However, we can observe that FA starts to assign more samples over time to the infrequent classes over time as it adapts the buffer slots based on the frequency of repetition. Moreover, it is evident in the plots that, by increasing the fraction of infrequent classes, the ratio gap between FA and CB gets smaller as the quota for CB stays the same while the number of infrequent classes increases. Eventually, when the fraction of infrequent classes is equal to $100\%$, i.e. all classes have the same (low) probably of repetition, all buffers have exactly the same ratio since all classes are infrequent.

In conclusion, FA buffer slots can be very helpful in highly unbalanced streams where a smaller fraction of classes have a low probability of repetition. When the stream moves towards becoming

balanced, the FA and CB get closer, and all methods become similar in the extreme case of a fully balanced stream with similar probability of repetition.

## F CHANGING $\mathcal{P}_f(\mathcal{S})$

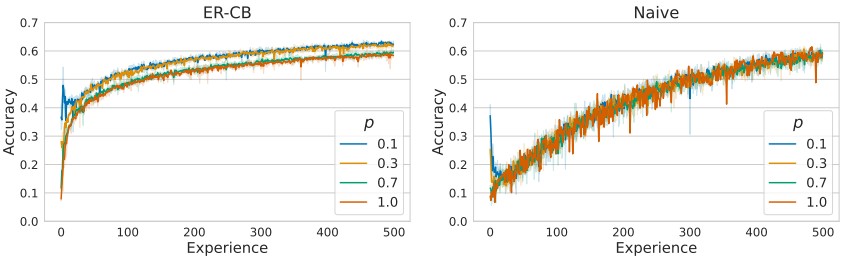

Figure 20: Average test accuracy for different values of $p$ in first occurrence.

We conduct experiments to find the differences between situations in which all classes occur early versus those in which new classes also appear late in the stream in order to analyze the role of first occurrence type. How early or late in the stream we observe all classes of a dataset, depends on the the parameters that control $\mathcal{P}_f(\mathcal{S})$. In this experiment, we fix the probability of repetition $P_r$ and change $\mathcal{P}_f(\mathcal{S})$'s parameters. In particular, we opt the geometric distribution for $\mathcal{P}_f(\mathcal{S})$ and choose the values $\{0.1, 0.3, 0.7, 1.0\}$ for its only parameter $0 < p \leq 1.0$. Increasing $p$ is inversely proportional to the spread factor in the first occurrence distribution, i.e. when $p$ is close to $0$ all classes happen in the first experience and as we move $p$ toward $1.0$ the classes start to spread along the stream. Figure 20 shows the CIFAR-100 results for the Naive and ER-CB strategies. The results suggest that when the spread factor is low, the model initially has difficulty to learn since there are more classes in the initial experiments and thus the model has to learn from fewer instances. However, with more experiences, all first occurrence types, reach almost the same SCA.

## G ER-FA RESULTS

Results in Figure 21 illustrate the total test accuracy and accuracy of frequent classes over time. Although the discrepancy between the accuracies of frequent classes is very small, the total test accuracy can significantly vary due to the difference in the accuracy of infrequent classes as presented in Section 4.4.

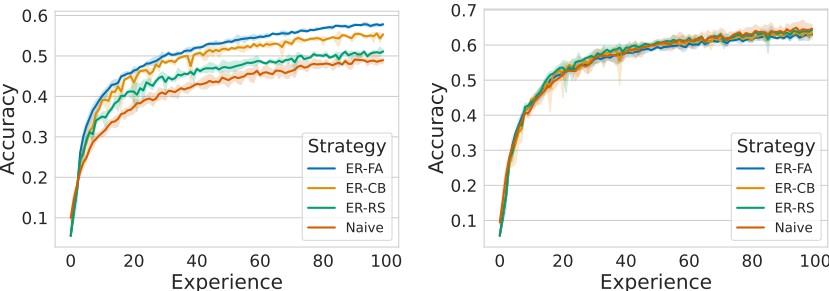

Figure 21: TA over all classes (left) and frequent classes (right) in a bi-modal unbalanced scenario with Fraction=0.3.

