# OpenReview forum: "Class-Incremental Learning with Repetition"
_ICLR.cc/2023/Conference — Submitted to ICLR 2023_

### Official Review · Reviewer_U14g · 2022-10-20

**Confidence:** 4
**Correctness:** 3
**Technical Novelty And Significance:** 2
**Empirical Novelty And Significance:** 3
**Recommendation:** 6

**Clarity, Quality, Novelty And Reproducibility:**

In terms of algorithms, the paper describes the steps of the algorithms in detail so that the algorithms can be clearly understood.

In terms of experimental results, the paper provides multiple experiments in various settings and uses high-quality graphs to present the experimental results.

In terms of novelty, the paper proposes two novel CLR generators and an improved FA method on ER method.

In terms of reproducibility, the author promises to provide an open-source implementation of our generators and algorithms with the scripts needed to reproduce the results reported in the paper.

In summary, the paper is well above average in terms of clarity, quality, novelty, and reproducibility.


**Strength And Weaknesses:**

Strengths: The paper proposes two practical CIR generators and performs a comprehensive series of comparative experiments of existing CL methods, which provides the CL area with a referenceable basis for further research.

Weaknesses: The analysis of the FA storage policy is not sufficient. In the paper, the FA method is only proposed based on ER. Can the FA method be applied to other CL methods? How does the ER-FA method perform when the dataset is not unbalanced? Does FA cause a decrease in class accuracy for frequent classes? The author may want to respond to the weakness the reviewer raised.


**Summary Of The Paper:**

For the possible repetition in Continual Learning, the paper makes contributes in the following three areas: (1) The author proposes two stochastic scenario generators that produce a wide range of CIR scenarios starting from a single dataset and a few control parameters. (2) The author conducts a comprehensive evaluation of repetition in CL by studying the behavior of existing CL strategies under different CIR scenarios. (3) The author presents a novel replay strategy that exploits repetition and counteracts the natural imbalance present in the stream.

**Summary Of The Review:**

Class-Incremental with Repetition (CIR) scenarios are important in realistic scenarios but have not been sufficiently studied so far. This paper proposes two practical CIR generators and performs a comprehensive series of comparative experiments of existing CL methods, which provides the CL area with a referenceable basis for further research. Therefore, it is recommended to accept this paper if the questions about the weaknesses can be answered.

---

> ### Author Response · Authors · 2022-11-15
> **Response to Reviewer U14g**
>
> We thank the reviewer for the encouraging feedback and questions. We answer your questions separately:
>
> > The analysis of the FA storage policy is not sufficient.
>
> We’ve added an analysis of FA policy by comparing it to CB and RS in Appendix E.2 in the new version of the paper. In the analysis, we show how the different policies behave under different levels of scenario imbalance. In the case of a balanced stream (where all classes have the same repetition probabilities), CB, RS, and FA behave similarly. We also present how RS can perform poorly in terms of the storage of samples from less frequent classes when the fraction of infrequent classes is very small, which would lead to a big drop in the accuracy of infrequent classes.
>
> > FA is only proposed based on ER. Can it be applied to other CL methods?
>
> Yes, the concept of frequency awareness may also be applied to any other strategy type, e.g., through scaling the relevance of the weights by the frequency of the classes in a regularization-based strategy. In this paper, we picked ER because it is recognized to be the most effective way to overcome forgetting in CI scenarios. Therefore, we first showed that the default ER storage policies might fail in some CIR scenarios, and then we improved upon ER using a frequency-aware storage policy. Obviously, ER-FA can also be combined with any other strategy in the form of hybrid strategies.
>
> > How does ER-FA perform with dataset is not unbalanced?
>
> If the stream is not unbalanced, FA will assign the same quota to each class in the dataset over time, and thus, it will turn into a class-balanced storage policy. Class-balanced policies work well under balanced scenarios, therefore the performance of FA will also remain similar to that of CB. We’ve shown this effect in Appendix E.2.
>
> > Does FA cause a decrease in frequent classes?
>
> The degradation is minimal. In Appendix G (Figure 21), we demonstrate the performance of frequent classes over time in all strategies. The accuracies of frequent classes are very similar, however, the total test accuracy over all classes can be significantly different among different strategies due to the gap between their infrequent class accuracies. A possible limitation would be in the case where the frequency of a particular class abruptly changes after some iterations. In that case, FA will require some time to adapt to the new frequency, causing a temporary decrease in performance. But eventually, it'll be able to recover.

---

### Official Review · Reviewer_MpJD · 2022-10-21

**Confidence:** 4
**Correctness:** 3
**Technical Novelty And Significance:** 3
**Empirical Novelty And Significance:** 3
**Recommendation:** 3

**Clarity, Quality, Novelty And Reproducibility:**

Overall, this paper is not high-quality work.
Clarity: This paper is not easy to follow. The figures in this paper don't help others understand the authors' motivation. For example, Figure 1 does not make sense in intuitively showing the difference between CIR and CI (or DI). Besides, the authors don't provide a visual explanation of the slot-based generator. The tedious conclusion part makes it difficult to grip the main contributions of this paper.

Novelty: This paper is not rich in novelty, and lacks relevant theoretical analyses. The main contribution of this paper, in my view, is to build a CIR data stream. Recently, analogous works with frequency-aware replay are not rare.

Reproducibility: The authors promise to provide open-source code and the algorithm.


**Strength And Weaknesses:**

Strengths
1. The idea of CIR is absorbing and reasonable. Because for realistic data streams, it is natural to occurrences previously seen classes in new coming samples. Further, they designed a paradigm to build a stream with repetition via the proposed generators.
Weaknesses

1. The novelty of this paper is insufficient. In my view, this paper conducted a sampling procedure via two generators. As for frequency-aware replay, its improvement is dealing with unbalanced data. What is neglected in this paper, however, is that there have been numerous works on adaptative store policy in continual learning.

2. The writing of this paper is not clear enough to follow. Firstly, the explanation of the discrepancy between CIR and traditional incremental learning (i.e., class- and domain-incremental) is not clear. Figure 1 replaced with actual objects may give a more intuitive understanding of the authors' motivation. Moreover, it is necessary to use a coherent algorithm to introduce the overall procedure, however, absent in the main text. Finally, the conclusion part is tedious and should be refined.

3. The lack of formulas in this paper makes comprehending have more barriers. For example, the generators build streams part lacks the complementary mathematical explanation.

4. The authors should provide comprehensive theoretical analyses of their method, in my view, which is necessary for solid work.


**Summary Of The Paper:**

This paper focuses on class-incremental with repetition (CIR), which ranges from class- and domain-incremental learning. For building streams and managing repetition over time, the authors propose two CIR generators (i.e., slot-based and sampling-based), which are from two aspects to generate data streams. Moreover, the authors propose a frequency-aware (FA) storage policy tailored to CIR scenarios. Further, the authors perform the first comprehensive evaluation on CIFAR 100 and TinyImageNet to compare other replay approaches in the CIR scenario, and then show the effectiveness of FA.


**Summary Of The Review:**

This is not a high-quality paper without a clear expression and lack of solid theoretical analyses, which could not stimulate other research works. So, I recommend rejection.

---

> ### Author Response · Authors · 2022-11-15
> **Response to Reviewer MpJD**
>
> We thank the reviewer for the constructive comments. We reply to the raised concerns individually:
>
> > The novelty of this paper is insufficient.
>
> We would like to highlight the novelty and contributions in our paper: 1) We designed two flexible CIR generators able to produce streams with repetition from any classification dataset. This directly enables and supports further research on the topic since the creation of CIR streams is a challenging and not yet standardized process. 2) We designed a novel continual learning strategy specifically designed to work in unbalanced CIR scenarios, showing that CIR indeed requires customized solutions. 3) Our empirical analysis provides the first quantitative assessment of the role of repetition and class balance for many different continual learning strategies. While some results may seem intuitive, we are the first to study these problems at a quantitative level. We believe that our proposed framework paves the way for future research into more practical continual learning scenarios.
>
> >What is neglected in this paper, however, is that there have been numerous works on adaptative store policy in continual learning.
>
> To the best of our knowledge, the majority of the related work focuses on manually designed scenarios in their experiments. In this work, we compare our ER-based strategy to the most commonly used ER storage policies, i.e. class-balanced and reservoir sampling, which are considered SOTA in class-incremental learning. We would appreciate any related work that we might have missed.
>
> >The writing of this paper is not clear enough to follow. Firstly, the explanation of the discrepancy between CIR and traditional incremental learning (i.e., class- and domain-incremental) is not clear. Figure 1 replaced with actual objects may give a more intuitive understanding of the authors' motivation.
>
> We appreciate the feedback. We have updated Figures 1 and 3 correspondingly and added a new table in the version of the paper (Table 1) that provides a formal comparison of scenario types and compares their properties. We hope the updated illustrations and definitions make the difference between CI, DI, and CIR more clear.
>
> >Moreover, it is necessary to use a coherent algorithm to introduce the overall procedure, however, absent in the main text.
>
> The algorithms for the generative procedure of CIR scenarios are explained in the main body of the paper (Sections 2.1 and 2.2). Similarly, the storage policy algorithm is explained in the text (Section 3). We provide further details in the appendix. This includes the formal definition of both generators (Appendix B and C) and their pseudo-code (Appendix B.1 and C.1). Due to the limited space and the large space occupied by the algorithms, we could only add them to the Appendix. We believe the content of the main text is sufficient to understand the key details of our implementation, while the Appendix provides a complete description. We will also release the Python code for both generators such that reproducibility is guaranteed.
>
> > Finally, the conclusion part is tedious and should be refined.
>
> We have revised the conclusion by focusing on the main contributions of our work for further clarification.
>
> > The lack of formulas in this paper makes comprehending have more barriers. For example, the generators build streams part lacks the complementary mathematical explanation.
>
> In section 2, we formulate the problem of continual learning mathematically, and we’ve added Table 1 In the new version of the paper, which formally describes the properties of each scenario type. For each generative procedure of CIR scenarios, we outline the formulation embedded within the text. Additionally, pseudo-code is provided in Appendix B and C.
>
> > The authors should provide a comprehensive theoretical analysis of their method. It is necessary of solid work
>
> As previously mentioned, the main goal of this paper is to propose CIR scenarios as a more natural setting for continual learning. We inspect and analyze CIR scenarios empirically as a first step through various experiments and show how different CL approaches might fail or succeed in such scenarios. That said, for the sampling-based generator (Appendix C), we also investigate stream generation under different probabilistic distribution configurations. This includes Zipf (Eq. 1), Poisson (Eq. 2), and Geometric (Eq. 3) distributions. For each distribution, we show how the parameter configuration impacts stream generation. We believe that our work is an initial empirical step towards more natural CL settings and hope the CL community will benefit.
>
> > Besides, the authors don't provide a visual explanation of the slot-based generator.
>
> Thank you for your suggestion. We have added the visualization for the overall steps of the slot-based generator to Appendix B.1 in Figure 12 in the new version of the paper.

---

### Official Review · Reviewer_rWm4 · 2022-10-30

**Confidence:** 2
**Correctness:** 4
**Technical Novelty And Significance:** 3
**Empirical Novelty And Significance:** 3
**Recommendation:** 8

**Clarity, Quality, Novelty And Reproducibility:**

The paper is well written and clear about the details, therefore I consider it's easily reproducible.

**Strength And Weaknesses:**

Strengths:
 - novelty of the analysis
 - relevant CL algorithms analysed under the different scenarios
 - well written paper

Weaknesses:
 - the results are not necessarily surprising; e.g. it is known that balancing classes in training batches benefits performance / optimization

**Summary Of The Paper:**

The paper proposes a study on a different flavor of continual learning scenarios than previously considered in the literature. Precisely, one in which data separation is not strict among concepts, and later tasks can revisit one of the previously introduced classes. The authors name this "class incremental with repetitions" (CIL).

The authors show that in these scenarios storing and replaying data is less important (which is not surprising), unless it corrects for imbalanced exposures to the different classes. Therefore the authors propose a sampling method which balances the class representation during training.

Experiments on the standard computer vision tasks show comparisons between domain incremental, class incremental and class incremental with repetitions.

**Summary Of The Review:**

The CL community would benefit from an analysis of a more natural stream of tasks than the extreme cases considered so far. Due to the clear exposure, interesting analysis and experiments I suggest the paper should be accepted.

---

> ### Author Response · Authors · 2022-11-15
> **Response to Reviewer rWm4**
>
> We thank the reviewer for the positive feedback highlighting the importance of studying more natural streams in continual learning.
>
> Regarding class balancing, while prior works have shown the importance of balanced classes in the batch training of neural networks, the impact of imbalance and repetition has not been studied quantitatively in a continual stream of experiences. With our proposed framework, it is possible to synthesize different types of unbalanced streams in a controlled way and study the behavior of existing strategies in those scenarios. Our framework, the proposed generators, and experimental results provide the foundation necessary to design more efficient strategies in more natural CL scenarios.

---

### Author Response · Authors · 2022-11-23
**Awaiting for reviewers' responses**

We thank all reviewers for their feedback, which contributed to improve the quality of our paper.
We believe we have answered the concerns raised in the reviews, and we revised our manuscript accordingly.
In particular, as requested by Reviewer MpJD, the paper now includes both a formal presentation of Class-Incremental with Repetition and a more intuitive representation of our proposed generators. The paper should be therefore much easier to understand in the updated version.
We encourage the reviewers to let us know whether their concerns have been addressed. Otherwise, we remain available to discuss any further issue we may have overlooked.

---

### Decision · Program_Chairs · 2023-01-20

**Decision:**

Reject

**Justification For Why Not Higher Score:**

The technical and theoretical contributions of the work are very marginal. The only significant contribution of the paper are the new IL scenarios with repetition, which is not significant enough to meet the bar for acceptance at ICLR.

**Justification For Why Not Lower Score:**

N/A

**Metareview: Summary, Strengths And Weaknesses:**

# Summary of Contribution

This paper describes an scenario for Continual Learning with Repetition (CIR), a scenario in which task distributions are not disjoint (which distinguishes it from the class-incremental scenario). Class-incremental and domain-incremental learning can be thought of as extreme cases of CIR. The authors propose two stream generators: The *slot* generator which does not repeat examples, and the *sampling* generator which instead allows repetition of samples. The authors additionally propose an approach to exemplar replay (Frequency-Away Replay) to address the CIL task. Experimental results are given on multiple benchmarks using the two proposed generators.

# Strengths

The main strength of the paper is its proposal of new scenarios for class-incremental learning which can be seen as a relaxation of the (somewhat rigid) protocols used in the majority of works on incremental learning. The experimental evaluation, and comparison with non-balancing baselines confirm the importance of taking repetition into consideration.

# Weaknesses

+ **Marginal Contribution**: The main contribution of the work is the definition of the two stream generators that can be used to generate novel scenarios for class-incremental learning with repetition. The proposed approach to CIL (Frequency-Aware Replay) is a very straightforward approach, and similar rebalancing approaches have been applied to class-incremental learning with long-tailed class distributions (and learning for class-imbalanced problems in general) and thus is only marginally novel.

+ **Clarity**: There are some details of the proposed scenarios and frequency-aware replay that are not precisely described in mathematical detail. Some aspects regarding clarity were addressed during the discussion phase.

+ **Lacking Theoretical Foundations and Insights**: The paper is basically an illustration of how to model incremental learning with non-disjoint tasks. As such, it does not provide any particularly insightful theoretical analysis of the *phenomenon* of incremental learning with repetition and how it must be addressed (beyond frequency-based sampling of exemplars).

# Summary

The significant contribution of this work lies in the definition of new class-incremental learning scenarios with repetition. Although appreciable from the perspective of continual learning in general, the contributions to the theory and practice of incremental learning (to CIR and CIL) are very small and not particularly novel. Rebalancing based on class frequencies is a very well-known approach, and this paper does not provide any new insights into the problem in the continual learning scenario.